# Cu Modified TiO$_2$ Catalyst for Electrochemical Reduction of Carbon Dioxide to Methane

**Akihiko Anzai [1], Ming-Han Liu [1], Kenjiro Ura [2], Tomohiro G. Noguchi [1], Akina Yoshizawa [1] , Kenichi Kato [3], Takeharu Sugiyama [4] and Miho Yamauchi [1,5,6,7,*]**

1   International Institute for Carbon-Neutral Energy Research (WPI-I2CNER), Kyushu University, 744 Motooka, Nishi-ku, Fukuoka 819-0395, Japan; anzai@i2cner.kyushu-u.ac.jp (A.A.); liu.ming-han.591@m.kyushu-u.ac.jp (M.-H.L.); noguchi@i2cner.kyushu-u.ac.jp (T.G.N.); yoshizawa@i2cner.kyushu-u.ac.jp (A.Y.)
2   Department of Chemistry, Graduate School of Science, Kyushu University, 744 Motooka, Nishi-ku, Fukuoka 819-0395, Japan; urajiro.ichi.21@outlook.jp
3   RIKEN SPring-8 Center, 1-1-1 Kouto, Sayo-cho, Sayo-gun 679-5148, Japan; katok@spring8.or.jp
4   Research Center for Synchrotron Light Applications, Kyushu University, 6-1 Kasuga-koen, Kasuga 816-8580, Japan; sugiyama@rcsla.kyushu-u.ac.jp
5   Institute for Materials Chemistry and Engineering (IMCE), Kyushu University, 744 Motooka, Nishi-ku, Fukuoka 819-0395, Japan
6   Advanced Institute for Materials Research (WPI-AIMR), Tohoku University, 2-1-1 Katahira, Aoba-ku, Sendai 980-8577, Japan
7   Research Center for Negative Emissions Technologies (K-Nets), Kyushu University, Motooka 744, Nishi-ku, Fukuoka 819-0395, Japan
*   Correspondence: yamauchi@ms.ifoc.kyushu-u.ac.jp

**Abstract:** Electrochemical reduction of CO$_2$ (ECO$_2$R) is gaining attention as a promising approach to store excess or intermittent electricity generated from renewable energies in the form of valuable chemicals such as CO, HCOOH, CH$_4$, and so on. Selective ECO$_2$R to CH$_4$ is a challenging target because the rate-determining step of CH$_4$ formation, namely CO* protonation, competes with hydrogen evolution reaction and the C–C coupling toward the production of longer-chain chemicals. Herein, a Cu-TiO$_2$ composite catalyst consisting of CuO$_x$ clusters or Cu nanoparticles (CuNPs), which are isolated on the TiO$_2$ grain surface, was synthesized using a one-pot solvothermal method and subsequent thermal treatment. The Cu-TiO$_2$ catalyst exhibited high selectivity for CH$_4$, and the ratio of FE for CH$_4$ to total FE for all products in ECO$_2$R reached 70%.

**Keywords:** electrochemical reduction of CO$_2$; Cu; TiO$_2$

## 1. Introduction

Electrochemical reduction of CO$_2$ (ECO$_2$R), which uses renewable electricity to produce fuels and chemical feedstocks from CO$_2$, has attracted attention not only as an eco-friendly material synthesis process but also as a novel method to store intermittent renewable electricity which is generated from renewable energies such as solar, wind power, and so on [1]. Among metal catalysts, Cu is known to exhibit the highest activity in ECO$_2$R [2] and also produces multi-carbon products such as hydrocarbons and alcohols. Meanwhile, the selective production of CH$_4$, which is widely used as a fuel, is a demanding task but has not been achieved to the desired level in ECO$_2$R employing active Cu catalysts.

CH$_4$ formation proceeds through the addition of 8 electrons and 8 protons to CO$_2$ (Equation (1)).

$$CO_2 + 8H^+ + 8e^- \rightarrow CH_4 + 2H_2O \tag{1}$$

In this process, CH$_4$ formation is thought to occur via *CHO formation by protonation of *CO [3]. The *CHO formation competes with both the C–C coupling of two *CO and hydrogen evolution reaction (HER), which significantly lowers selectivity for the production

of $CH_4$. Recently, the formation of isolated Cu sites has been found to effectively improve the selectivity for the $CH_4$ production by suppressing the unfavorable C–C coupling [4,5]. However, the selectivity for the production of $CH_4$ from $CO_2$ ($CO_2$ to $CH_4$ selectivity) should be improved. The increase of surface area is found to efficiently suppresses the Cu agglomeration on a support material such as carbons [6] and oxides [4,7,8]. However, the formation relatively large Cu portions where C–C couplings preferentially progress cannot be suppressed in conventional impregnation synthesis.

Titanium dioxide ($TiO_2$) is a multifunctional material with many advantages, being ubiquitous, low-cost, and environmentally friendly. $TiO_2$ is a widely used catalyst material for various application [9] due to its high activity and adequate stability. Recently, we have uncovered relatively high overpotentials for electrochemical $H_2$ evolution and the favorable interactions of oxygen species included in organic acids, oximes, and imines with the surface of $TiO_2$-based catalysts for electrochemical hydrogenation [10–18], which could enhance the selectivity for the production of $CO_2$-derived chemicals in E$CO_2$R. Highly dispersed and isolated Cu sites on $TiO_2$ are expected to suppress the generation of multi-carbon products and show high selectivity for $CH_4$ synthesis. Thus, we develop Cu-$TiO_2$ catalysts presenting isolated Cu sites with high dispersion for the selective $CH_4$ production by the electrochemical reduction of $CO_2$.

## 2. Results

To enhance their dispersivity, Cu-$TiO_2$ catalysts were prepared via a one-pot solvother-mal method (See Section 4). The crystal structures of the prepared $TiO_2$, Cu-$TiO_2$, and Cu-$TiO_2$ treated with $H_2$ (Cu-$TiO_2$-H) were examined by X-ray diffraction (XRD) (Figure 1 and Figure S1). All samples showed a diffraction pattern mostly attributable to an anatase phase of $TiO_2$, but a very small peak from a brookite phase was also observed at 14.7°. The intensity of the brookite peak of Cu-$TiO_2$ seemed slightly large compared to that in the XRD pattern of $TiO_2$. It has been reported that the formation of brookite phase preferentially occurs in alkaline conditions. The usage of DMF as an alkalescent organic solvent prob-ably induces the formation of brookite phase [19–21]. No diffraction peaks from copper oxides such as $Cu_2O$ or CuO were observed in the Cu-$TiO_2$, despite the introduction of Cu. In contrast, Cu-$TiO_2$-H showed a diffraction peak from Cu. It is notable that no shift in the diffraction peak position of the anatase phase was observed in either Cu-$TiO_2$ or Cu-$TiO_2$-H, although the ionic radii of the 6-coordinated $Ti^{4+}$ ions (0.605 Å) and $Cu^{2+}$ ions (0.73 Å) are different and a $Cu^{2+}$ ion is larger than that of a $Ti^{4+}$ ion [22], which suggests that most of the Cu species are precipitated over the $TiO_2$ surface as $CuO_x$ clusters on Cu-$TiO_2$, where the clusters are too small to be detected by XRD, and as Cu nanoparticles (NPs) on Cu-$TiO_2$-H, rather than being enclosed in the $TiO_2$ lattice. To obtain detailed structural parameters for the catalysts, we conducted Rietveld profile fitting of these XRD patterns (Figure S1). Structural parameters are summarized in Table S1. The lattice constants of anatase phase constituting Cu-$TiO_2$ and Cu-$TiO_2$-H showed a slight increase in the *a*-axis but a slight decrease in the *c*-axis compared to those of pure $TiO_2$, suggesting the possibility of incorporation of Cu species into the $TiO_2$ lattice. The weight fraction of Cu species deposited on $TiO_2$ was estimated by the Rietveld analysis to be low (1.9%), even though the initial starting amount was 10%. Considering the slight change in the structure of Cu-$TiO_2$ and Cu-$TiO_2$-H, a relatively large percentage of Cu species (more than 5%) possibly exists as amorphous on the surface of $TiO_2$.

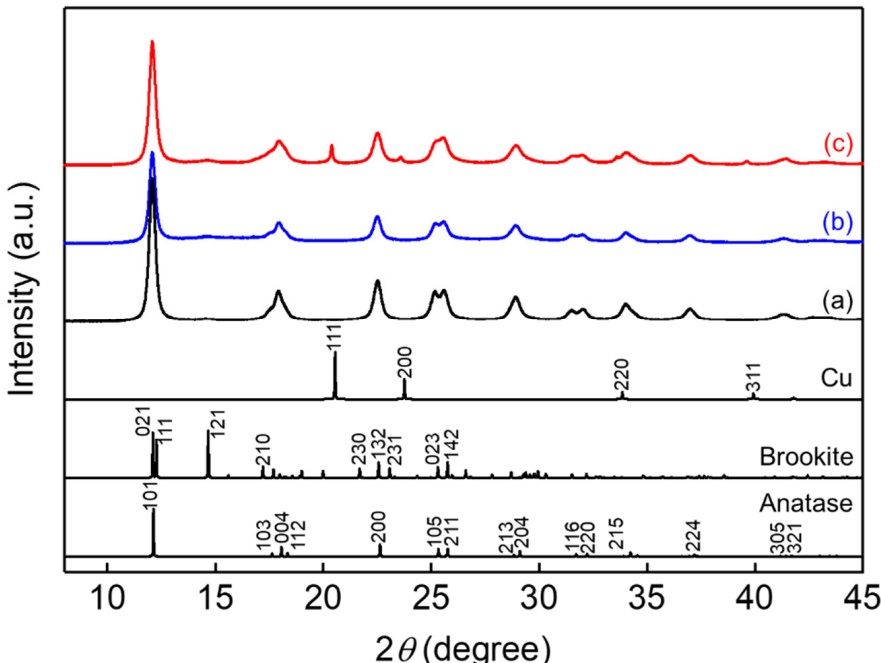

**Figure 1.** XRD patterns of (**a**) $TiO_2$, (**b**) $Cu-TiO_2$, and (**c**) $Cu-TiO_2$-H. The simulated XRD patterns of anatase, brookite, and Cu are also presented.

To confirm the oxidation state of Cu on $Cu/TiO_2$, the diffuse reflectance UV-Vis spectra of the catalyst were measured (Figure 2). The inset represents the Tauc plots [23,24] where the $(F(R) hv)^{1/2}$ is plotted as a function of photon energy. The indirect band gap was estimated from an X-intercept of a linear fit of the Tauc plot, and the determined band gap values are also given in the inset. The band gap energy of $TiO_2$ was estimated to be 3.10 eV, which is slightly small compared to the reported value of anatase $TiO_2$ [25]. $Cu-TiO_2$ showed a large red-shift in optical absorption edge compared to that of $TiO_2$, with an estimated band gap of 1.46 eV, which occurs probably by interaction between CuO and $TiO_2$. Furthermore, absorption bands appeared in the region of 400–600 nm and 600–1200 nm. The former absorption band is assignable to the interfacial charge transfer from a $TiO_2$ O 2p valence band to a Cu(II) ions connected to $TiO_2$ [26], whereas the latter comes from a d–d transition of Cu (II) species [27]. These Cu(II) states may exist either as Cu(II) clusters or amorphous oxide grains of CuO. $Cu-TiO_2$-H exhibited a remarkably larger red-shift of the absorption edge in optical absorption than $Cu-TiO_2$ and with 1.16 eV of band gap, revealing good contact between Cu and $TiO_2$ grains. Additional absorption bands appeared in the region of 400–600 nm and 550–1200 nm. The former absorption band was assigned to the interfacial charge transfer from the $TiO_2$ O 2p valence band to the Cu(II) ions attached to $TiO_2$ [26], whereas the latter band was attributable to the Cu surface plasmon resonance of Cu NPs [28–30]. Therefore, we confirmed the formation of Cu NPs in these catalysts from the XRD and DRS results.

To investigate the dispersion of Cu NPs on $TiO_2$, we conducted scanning transmission electron micrography (STEM) for $Cu-TiO_2$-H. A high-angle annular dark-field STEM (HAADF-STEM) image of the $Cu-TiO_2$-H (Figure 3) suggested that Cu NPs with a diameter of 2–3 nm, which appear as white dot-like objects, were well dispersed over a $TiO_2$ grain, indicating that isolated Cu clusters are formed in $Cu-TiO_2$-H. On the other hand, $Cu-TiO_2$ (Figure S2) suggested that Cu NPs with a diameter of 2–5 nm, which appear as white aggregated objects, existed in a $TiO_2$ grain, indicating that larger Cu NPs are formed in $Cu-TiO_2$ compared to $Cu-TiO_2$-H.

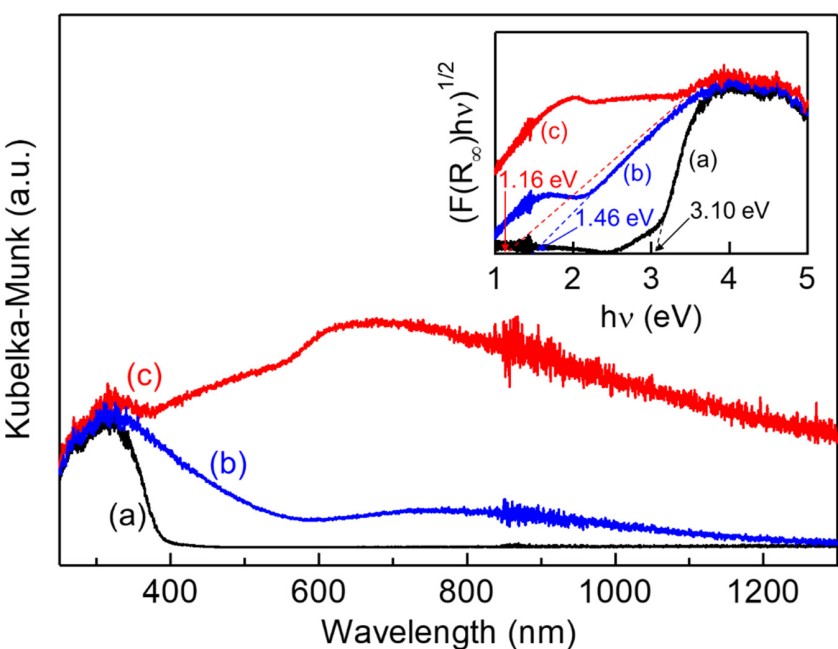

**Figure 2.** Diffuse reflectance spectra of (**a**) TiO$_2$, (**b**) Cu-TiO$_2$, and (**c**) Cu-TiO$_2$-H samples. The inset is the Tauc plots for the determination of the bandgap of the samples.

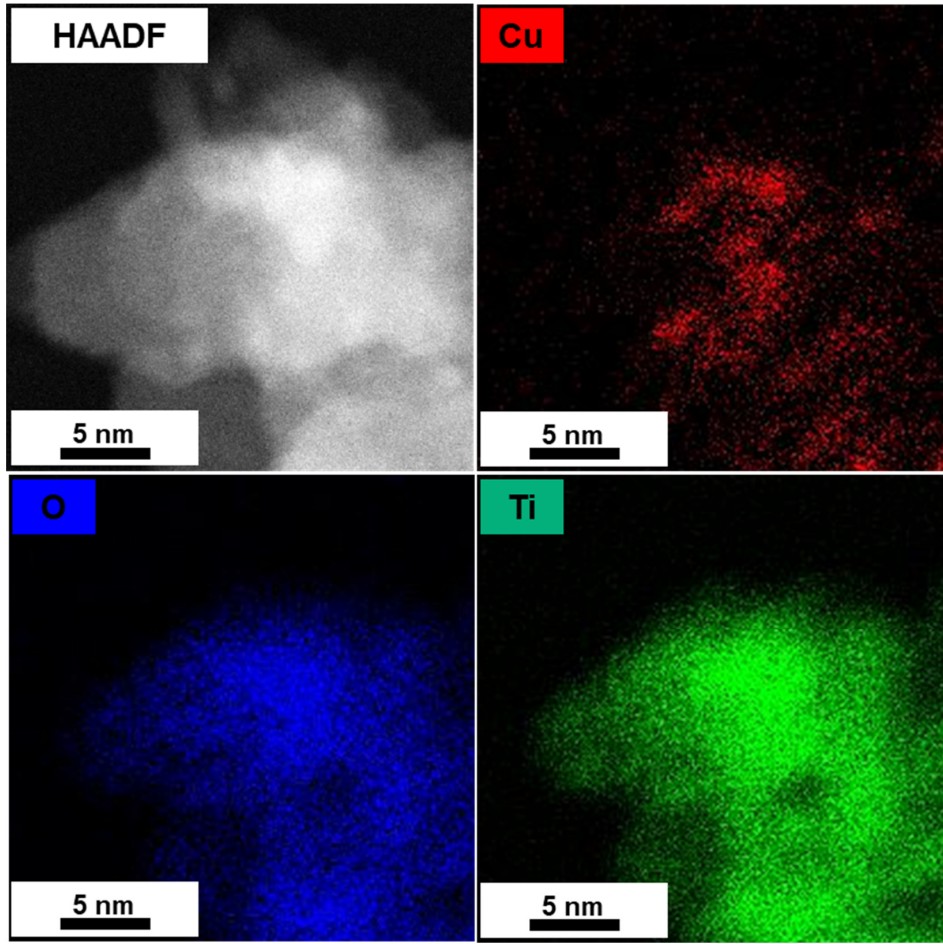

**Figure 3.** High-angle annular dark-field scanning transmission electron microscopy (HAADF-STEM) image and EDS mapping images of Cu-TiO$_2$-H.

Figure 4 provides Cu $2p_{3/2}$ XPS spectra of Cu-TiO$_2$ and Cu-TiO$_2$-H. A relatively broad Cu 2p3/2 XPS peak centered at 932.5 eV and the shoulder peak near 934.3 eV in the high binding energy side were observed on Cu-TiO$_2$. The main and the shoulder peaks could be attributed to Copper oxide containing Cu(I) and Cu(II), and Cu(OH)$_2$ [31] species, respectively, which implies that Cu clusters in Cu-TiO$_2$ are composed of Cu(I) and Cu(II). Furthermore, the presence of a well known shake-up satellite at higher binding energies than that for the main peak strongly indicates the presence of Cu(II) species in Cu-TiO$_2$ [32]. Therefore, Cu clusters in Cu-TiO$_2$ may exist mainly as CuO-like and Cu(OH)$_2$-like species. Cu-TiO$_2$-H showed the sharp Cu 2p3/2 peak centered at 932.6 eV, which is attributed to the formation of Cu(0). The absence of the shake-up satellite, as observed in Cu-TiO$_2$, suggests that Cu NPs on Cu-TiO$_2$-H are mainly composed of Cu(0) species, which is consistent with the observation in the XRD measurement for Cu-TiO$_2$-H. Figure S3A shows Ti 2p XPS spectra of TiO$_2$, Cu-TiO$_2$, and Cu-TiO$_2$-H. All catalysts exhibited a symmetrical peak centered around 458.6 eV, which is a typical peak with characteristic binding energy value for Ti$^{4+}$ ions contained in anatase TiO$_2$ [33], although the spectrum of TiO$_2$ had a slightly extended tail which possibly comes from the formation of Ti$^{3+}$. There was also no obvious peak shift in these catalysts. Therefore, Ti ions near the surface of all catalysts would have an analogous chemical environment. Figure S3B represents O1s peaks at around 529.7 eV in XPS spectra of TiO$_2$, Cu-TiO$_2$, and Cu-TiO$_2$-H, which are assigned to lattice oxygen of TiO$_2$ [33]. In addition, each spectrum contained two other weak shoulder peaks on the higher binding energy side of the main O 1s peak. The former peak observed at 530.8 eV can be attributed to the hydroxide group or water molecules that are present at the surface [34], and the other peak at 531.9 eV is originated from organic contaminants containing oxygen species [35]. Cu-TiO$_2$-H showed a relatively large peak at 530.8 eV, which is attributed to the existence of hydroxide groups or water molecules. Chalastara et al. reported that a brookite-rich sample has a larger amount of surface-bound OH/H$_2$O groups than anatase-rich samples [36]. The increase of intensity at 530.8 eV indicates the increase of a brookite phase in Cu-TiO$_2$-H. The number of defects such as Ti$^{3+}$ and oxygen vacancy on the surface of these catalysts did not change so much by the introduction of Cu or by the change of the atmosphere during heating. The all-measured core level positions for the samples are summarized in Table S2.

The difference in Cu state between Cu-TiO$_2$ and Cu-TiO$_2$-H was further examined by the measurement of Cu-K edge X-ray absorption near edge structure (XANES) spectra using a conversion electron yield (CEY) method, which reflects information on the species near surface region due to the shallow escape depth of Auger electrons from an atom irradiated by X-rays within a sample [37]. Figure 5 shows XANES spectra of Cu-TiO$_2$ and Cu-TiO$_2$-H, and standard samples such as Cu foil, Cu$_2$O, and CuO. Small pre-edge bumps at 8978 eV were observed on Cu-TiO$_2$ and CuO, which is attributed to dipole-forbidden 1s → 3d transition and is indicative of existence of Cu$^{2+}$ ions, implying that the surface of Cu-TiO$_2$ contains Cu(II) ions. Cu-TiO$_2$-H showed an XANES spectrum similar to that of Cu foil, suggesting that the Cu species on the surface of Cu-TiO$_2$-H are Cu(0) species. XANES spectra were further analyzed in detail. The first derivatives of the XANES spectrum of CuO gave a small pre-edge bump at 8978 eV for the 1s → 3d transition [38], peak for the 1s → 4p transition [39,40] (Figure 5). The spectrum also showed a small pre-edge bump at 8978 eV attributed to the 1s → 3d transition, but the spectral features in the region related to the 1s → 4p transition were significantly different from those of CuO, meaning that the oxidation state of Cu on Cu-TiO$_2$ is different from that of CuO, although it includes Cu(II) species. The red shift and increase in intensity of the 1s → 4p peak was interpreted as an increase of covalency in the ligand−copper bond [40]. Cu(II) states on Cu-TiO$_2$ may emerge as Cu(OH)$_2$-like species, which is consistent with the observation in the XPS measurement for Cu-TiO$_2$. The first derivative spectra of Cu-TiO$_2$-H and Cu foil look similar, indicating that Cu(0) is the main oxidation state of Cu species in Cu-TiO$_2$-H, which is consistent with all observations discussed above.

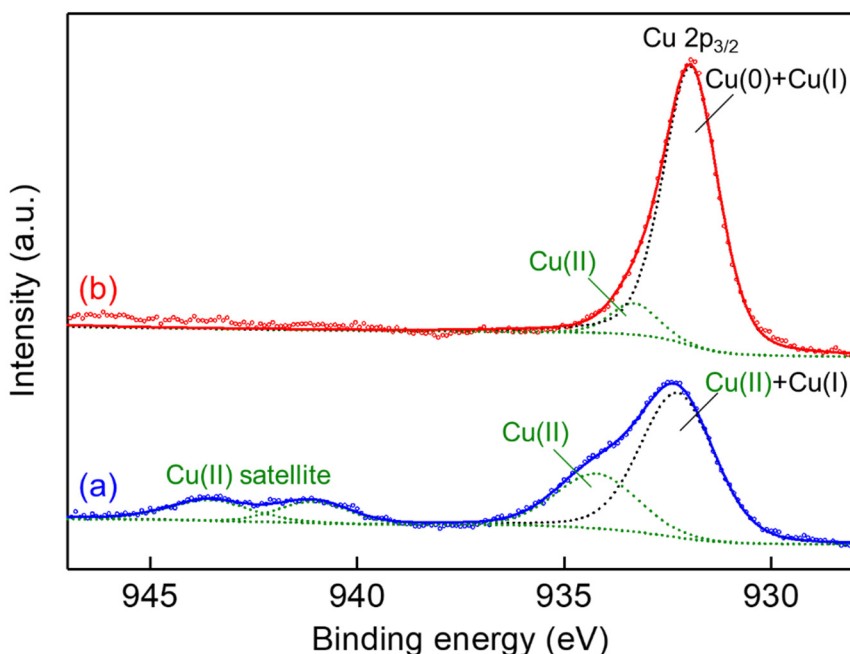

**Figure 4.** Deconvoluted Cu 2p$_{3/2}$ XPS spectra of Cu-TiO$_2$ (**a**) and Cu-TiO$_2$-H (**b**). The observed spectra, fitting curves, calculated patterns, and deconvoluted curves are denoted by circle, solid line, and dashed line, respectively.

Figure 6A shows linear sweep voltammetry (LSV) curves under CO$_2$ flow for TiO$_2$, Cu-TiO$_2$, and Cu-TiO$_2$-H. Cu-TiO$_2$-H exhibited a higher current density than the other two catalysts, meaning that the formation of Cu NPs on TiO$_2$ increases ECO$_2$R activity. All catalysts produced H$_2$, CO, CH$_4$, and C$_2$H$_4$ as gaseous products, which were detected by online gas chromatography. We could not observe other gaseous product such as HCOOH. Only HCOO$^-$ was detected in the liquid phase by HPLC analysis, as shown in Figures S4–S6. TiO$_2$ mainly produced H$_2$ via the hydrogen evolution reaction (HER), and a small amount of HCOO$^-$, CO, and CH$_4$ were also produced, suggesting that TiO$_2$ does not show high activity for ECO$_2$R. On Cu-TiO$_2$, the percentage of CH$_4$ in the products increased and a tiny amount of C$_2$H$_4$ was also produced. This suggests that Cu(II) species are reduced to Cu(0) in Cu NPs under the potential and enhances ECO$_2$R activity. The measured total Faradaic efficiencies on the TiO$_2$ and Cu-TiO$_2$ sometimes became slightly higher than 100%, which has been explained by the experimental errors introduced by GC detection or inconsistencies in the flow rate, as shown in some reports [41–43]. Cu-TiO$_2$-H exhibited higher ECO$_2$R activity than TiO$_2$ and Cu-TiO$_2$, as shown in the LSV results. Faradaic efficiency for the production of CH$_4$ reached 18% with 36 mA cm$^{-2}$ of partial current density at −1.8 V vs. RHE (Figure 6F), where CH$_4$ partial current density was defined as a product of the average total current density and the Faradaic efficiency for the production of CH$_4$ in ECO$_2$R at each potential. Notably, Faradaic efficiency for CH$_4$ production much increased at potentials more negative than −1.4 V, indicating that CO$_2$ is selectively converted to CH$_4$ with the applied potentials. To evaluate the selectivity for CH$_4$ formation, we further calculated the ratios of FE for CH$_4$ to total FE for all products in ECO$_2$R (FE$_{CH4}$/FE$_{C1+C2}$) on the catalysts at each potential (Figure 6E). Cu-TiO$_2$ and Cu-TiO$_2$-H showed larger FE$_{CH4}$/FE$_{C1+C2}$ values than TiO$_2$. This result is probably attributable to Cu sites homogeneously dispersed on these catalysts. We achieved 70% of FE$_{CH4}$/FE$_{C1+C2}$ at −1.8 V vs. RHE, which compares with prior reports related to highly dispersed or single-site copper catalysts (Table S3). On the other hand, Cu-TiO$_2$ and Cu-TiO$_2$-H showed different CH$_4$ partial current densities (Figure 6F (b) and (c)), although the selectivity for CH$_4$ in the products from CO$_2$, i.e., FE$_{CH4}$/FE$_{C1+C2}$ over these catalysts looked similar (Figure 6E (b) and (c)). DRS results represented that the band gap of Cu-TiO$_2$-H is narrower

than that of Cu-TiO$_2$ (Figure 2), which implies that Cu species on Cu-TiO$_2$-H have better contact with the TiO$_2$ support than those on Cu-TiO$_2$. Such favorable interactions between Cu catalysts and the TiO$_2$ support in Cu-TiO$_2$-H may enhance the electrical conductivity, resulting in its high CH$_4$ partial current density on Cu-TiO$_2$-H. These results indicate that Cu(0) loading on TiO$_2$ is indispensable and effective to enhance activity in the ECO$_2$R to CH$_4$. However, the state of catalysts under operating conditions are not well clarified and need to be further studied.

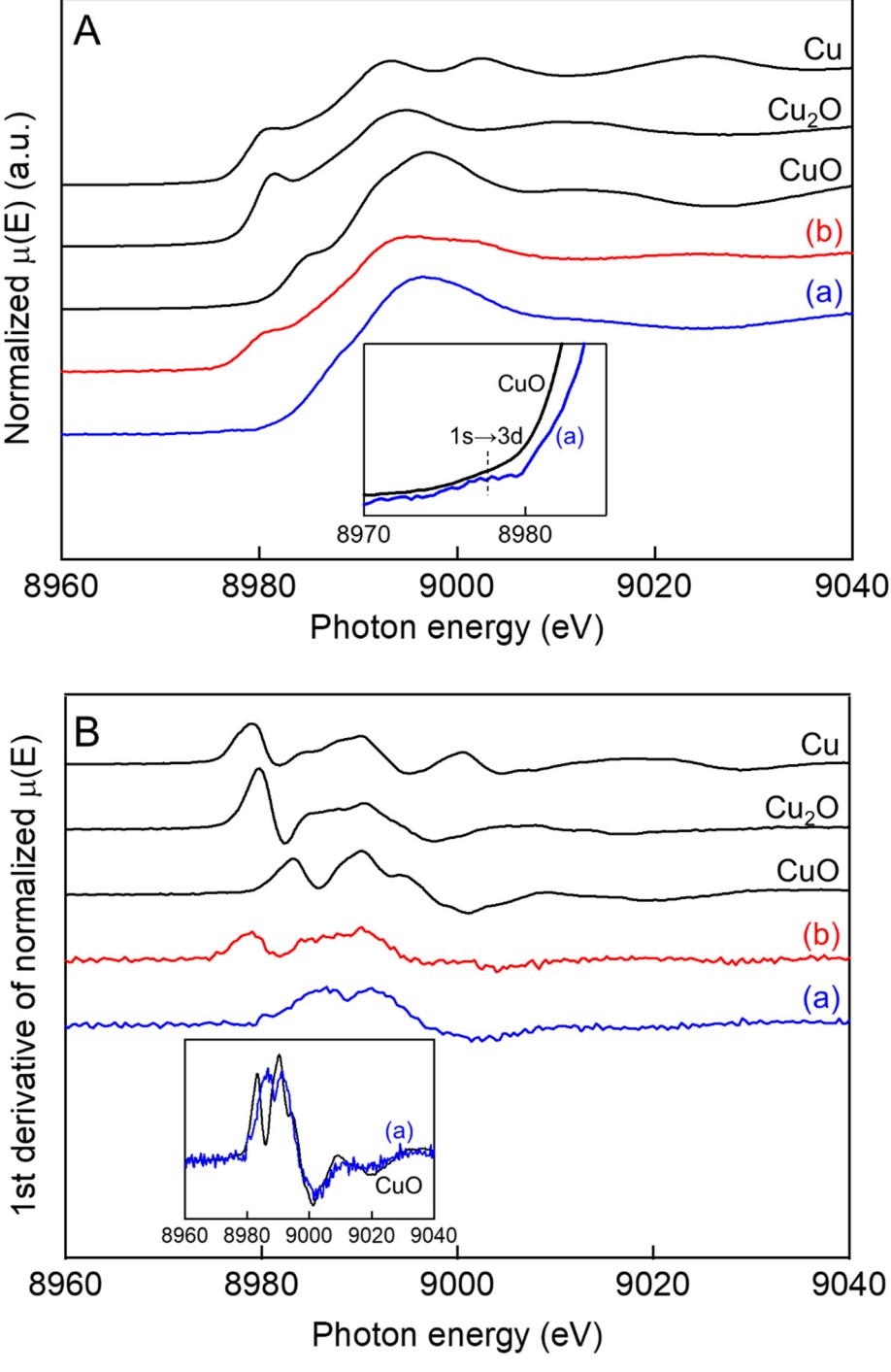

**Figure 5.** Cu K-edge spectra (**A**) and first derivative of Cu K-edge XANES spectra (**B**) of reference compounds. (a) Cu-TiO$_2$, (b) Cu-TiO$_2$-H. The inset figure shows overlay of the spectra of CuO and (a) Cu-TiO$_2$.

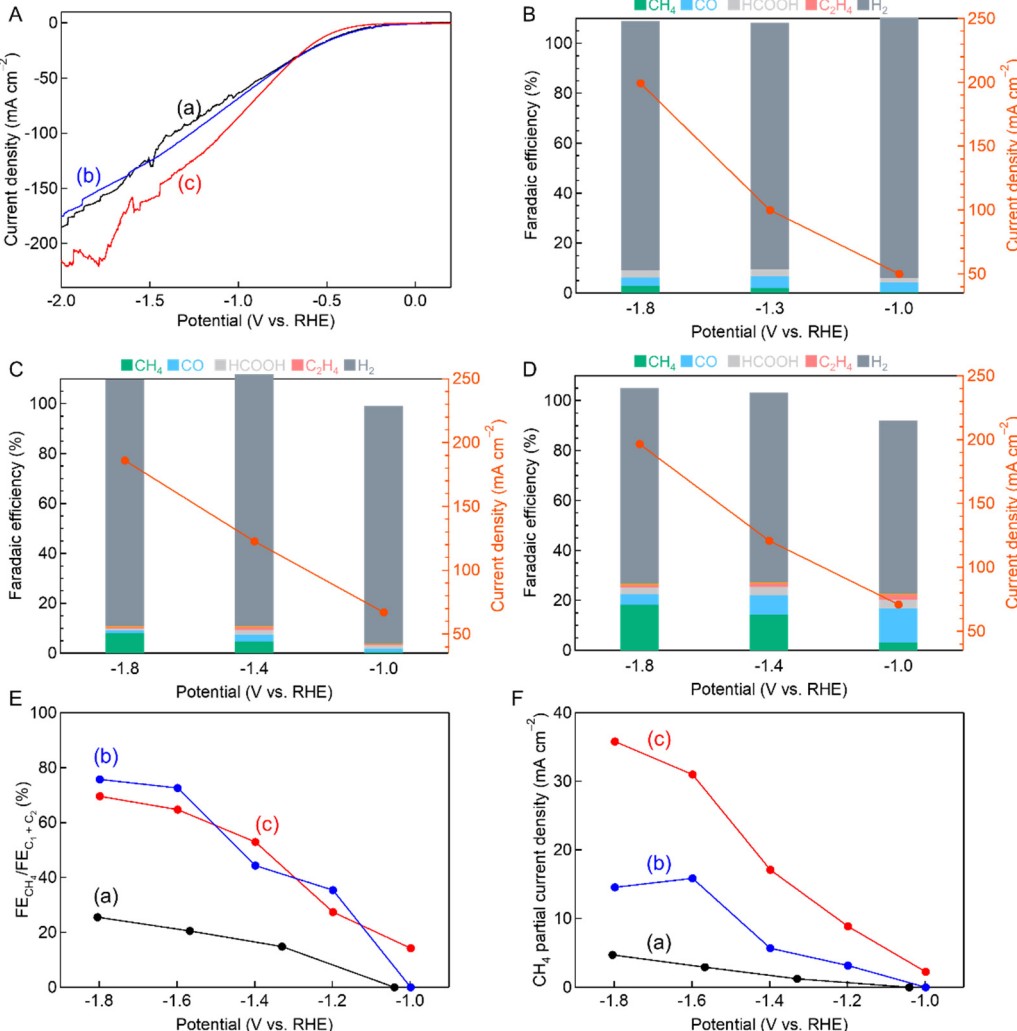

**Figure 6.** LSV curves under $CO_2$ flow (**A**) on an electrode employing (a) $TiO_2$, (b) $Cu$-$TiO_2$, and (c) $Cu$-$TiO_2$-H catalyst. FEs for different products over $TiO_2$ (**B**), $Cu$-$TiO_2$ (**C**), and $Cu$-$TiO_2$-H (**D**) catalysts at various potentials. Comparison of $FE_{CH4}/FE_{C1+C2}$ (**E**) on (a) $TiO_2$, (b) $Cu$-$TiO_2$, and (c) $Cu$-$TiO_2$-H catalyst. Comparison of $CH_4$ partial current density (**F**) of (a) $TiO_2$, (b) $Cu$-$TiO_2$, and (c) $Cu$-$TiO_2$-H catalyst.

It has been reported that composite materials of Cu and $CeO_2$ exhibit high $CH_4$ selectivity [4,7,44,45], where abundant oxygen vacancy ($V_O$) sites on $CeO_2$ play an important role in both $CO_2$ adsorption and activation. Wang et al. suggested that Cu site surrounded by $3V_O$ effectively promoted $CH_4$ formation, where the $2V_O$ neighboring Cu are filled with the two oxygen atoms of $CO_2$ and should enhance the $CO_2$ adsorption and activation [4]. In this study, the surface of the prepared $TiO_2$ samples may consist of low concentration of $V_O$. $CO_2$ dissociation unfavorably occur on stoichiometric anatase surface compared to Vo-rich surface due to its large $CO_2$ dissociation energy [46]. HER mainly proceeded on the $TiO_2$ catalyst, although small amounts of C1 products, such as CO, $HCOO^-$, and $CH_4$, were generated (Figure 6B). In contrast, $Cu$-$TiO_2$ and $Cu$-$TiO_2$-H, where Cu(0) species were well dispersed over a $TiO_2$ grain, showed high selectively for the production of $CH_4$. Hence, well dispersed Cu species on $TiO_2$ would play a key role in the formation of $CH_4$. Therefore, it is reasonable that homogeneity of Cu NPs in their size and dispersity on $TiO_2$ can be a key for the high $CO_2$-to-$CH_4$ selectivity.

## 3. Materials and Methods

### 3.1. Materials

Titanium tetrabutoxide monomer (95.0%), anhydrous copper (II) acetate (97%), N,N-dimethylformamide (99.5%), 2-propanol (99.7%), ethanol (99.5%), acetone (95.5%), hexane (96.0%), and potassium hydroxide were purchased from FUJIFILM Wako Pure Chemical Corporation. Nafion® perfluorinated resin solution (5 wt% in mixture of lower aliphatic alcohols with a water content of 45%) was purchased from Sigma-Aldrich Corporation. All chemicals were used without further purification. All the solutions were prepared with deionized water.

### 3.2. Preparation of Cu-TiO$_2$ Composite Catalysts

For the preparation of Cu-TiO$_2$ composite catalyst, including 10 wt% of Cu species, titanium tetrabutoxide (2.8 mmol, 1 mL) was quickly added to a mixture of 30 mL of N,N-dimethylformamide (99.5%), 0.215 mL of 2-propanol, and 73 mg of anhydrous copper(II) acetate in a 50 mL Teflon-lined stainless-steel autoclave to avoid exposing the sample to the air, and the mixture was sonicated for 2 h at room temperature. The container was sealed, heated from room temperature to 200 °C over 30 min in an electrical oven, and maintained for 20 h. The product was separated by centrifugation 7500 rpm for 10 min and washed several times with ethanol, acetone, and hexane. After that, it was dried under vacuum at room temperature. Finally, the precursor was calcined at 450 °C for 30 min under air or flowing H$_2$ (60 mL min$^{-1}$) to obtain TiO$_2$ and Cu-TiO$_2$-x samples, where X is the calcination atmosphere, air or H$_2$; heating rate of 15 °C min$^{-1}$.

### 3.3. Catalyst Characterization

The metal composition of prepared oxides was determined using an energy dispersive X-ray spectroscope (EDS, JED-2300, JEOL) equipped with the SEM instrument. Powder X-ray diffraction (XRD) patterns were obtained using synchrotron radiation (λ = 0.740040 Å) at RIKEN Materials Science beamline BL44B2, SPring-8 [47]. Data were acquired using the high-resolution Debye–Scherrer camera equipped with an imaging plate as an X-ray detector. Rietveld analyses were performed using a Topas software package (Bruker AXS Inc., Billerica, MA, USA, version 5). The diffuse reflectance spectra of samples were recorded using a V-670 spectrometer (JASCO, Japan) equipped with an integrating sphere. The diffuse reflection spectra were converted into reflectance spectra using the Kubelka-Munk function. X-ray photoelectron spectroscopy (XPS) studies were performed on a VersaProbeII (ULVAC-PHI) using nonmonochromatic Al Ka radiation. Binding energies in XPS spectra were corrected by referring a C 1s binding energy of the carbon atoms of the ligand in the specimens at 284.6 eV. Scanning transmission electron microscopy (STEM) image was obtained using a JEM-ARM200F (JEOL Co., Tokyo, Japan) at Kyushu University operated at 200 kV. Sample grids for the STEM observations were prepared by dropping ethanol dispersions of the specimens onto a carbon-supported nickel grid. Cu K-edge X-ray absorption fine structure (XAFS spectra of the samples) was measured at Kyushu University beamline BL06 of Kyushu Synchrotron Light Research Center (SAGA-LS, Japan) with an electron storage ring operating at the energy of 1.4 GeV. The energy range of this light source (bending magnet) is 2.1–23 keV. A silicon (111) double-crystal monochromator was used to obtain the incident X-ray beam. The typical photon flux is 10$^{10}$ photons per second. The spectra of standard samples such as Cu foil, Cu$_2$O, and CuO were recorded in the transmission mode at 20 °C using a Si(111) double-crystal monochromator. The spectra of Cu-TiO$_2$ samples were measured using the conversion electron yield mode. Data processing was carried out by Athena and Artemis included in the Ifeffit package [48].

### 3.4. Electrode Preparation

The cathode GDE was prepared by airbrushing catalyst inks onto a gas diffusion carbon paper (Fuel Cell Store Sigracet 22 BB, with a microporous layer) with a carrier gas of air. The catalyst ink was prepared with 200 μL of 2-propanol, 200 μL of water,

10 μL of Nafion® solution, and 1 mg of catalyst powder. The catalyst ink mixtures were sonicated in a 4 mL screw neck glass vial for 15 min, and then sprayed onto the gas diffusion carbon paper.

### 3.5. Electrochemical Reduction of $CO_2$

The electrochemical measurements were conducted in an electrochemical flow cell setup configuration with the three-electrode system. The geometric area of the cathode in the flow cell is $1\,cm^2$, which is used for all current density calculations. 1 M KOH aqueous solution was introduced into the cathode chamber at the rate of $7\,mL\,min^{-1}$ and the anode chamber at the rate of $1\,mL\,min^{-1}$ by two pumps, respectively. A Nafion 117 cation exchange membrane (Chemours®) was used to separate the cathode chamber and anode chamber. Pure $CO_2$ gas (Linde, 99.99%) was continuously supplied to the gas chamber of the flow cell at a flow rate of $15\,mL\,min^{-1}$. The $CO_2RR$ performance was tested using constant-current electrolysis, i.e., chronopotentiometry while purging $CO_2$ into the catholyte during the whole electrochemical test. The potentials vs. Hg/HgO reference electrode were converted to values vs. reversible hydrogen electrode (RHE) using the following equation [49].

$$E \text{ (vs. RHE)} = E \text{ (vs. Hg/HgO)} + 0.098 \text{ V} + 0.0591 \text{ V} \times \text{pH} \tag{2}$$

All voltages reported are without $iR$ compensation.

Gas products were analyzed by on-line gas chromatography (Micro GC Fusion®, Inficon, Bad Ragaz, Switzerland) with a Molsieve 5A column and a Plot Q column coupled with thermal conductivity detector (TCD). Liquid products were analyzed high performance liquid chromatograph (HPLC, LC-20AD, Shimadzu) equipped with a refractive-index detector (RID-10A, Shimadzu). The Faradaic efficiency (FE) of products in the electroreduction experiments is defined by the following equation:

$$FE_i = \frac{n_i \times z_i \times F}{Q} \times 100 \tag{3}$$

where $n_i$ is the number of moles of product $i$, and $z_i$ represents the number of electrons required for the formation of product $i$ ($z_i = 2$ for CO, formic acid, and $H_2$; $z_i = 8$ for $CH_4$; $z_i = 12$ for $C_2H_4$; $z_i = 14$ for $C_2H_6$). $F$ is the Faraday constant (96,485 C $mol^{-1}$ of electrons). $Q$ is the amount of charge passed during the electrolysis. For the gas products, $n_i$ was calcurated as follows:

$$n_{i,\text{gas}} = \frac{P_0 \times x_i \times v \times t}{R \times T} \tag{4}$$

where $x_i$ is the volume fraction of gas product $i$; $P_0$ is atmospheric pressure (1 atm); $v$ is the $CO_2$ flow rate (0.015 L $min^{-1}$); $t$ is electrolysis time; $R$ is the ideal gas constant (0.08205 L atm $mol^{-1}$ $K^{-1}$); $T$ is 298 K.

## 4. Conclusions

We successfully prepared Cu-$TiO_2$ composite catalysts, where Cu clusters or NPs were well dispersed by a one-pot solvothermal method and subsequent thermal treatment for electrochemical reduction for $CO_2$. For Cu-$TiO_2$ sample obtained by calcination of the precursor in air, $CuO_x$ cluster were dispersed on $TiO_2$ surface and Cu NPs were formed on Cu-$TiO_2$-H obtained by hydrogen treatment of the precursor. Cu-$TiO_2$-H was found to exhibit high selectivity for $CH_4$ in $ECO_2R$. Faradaic efficiency for the $CH_4$ production reached 18% with a $CH_4$ partial current density of $36\,mA\,cm^{-2}$ at $-1.8$ V vs. RHE. Furthermore, 70% of $FE_{CH4}/FE_{C1+C2}$ at $-1.8$ V vs. RHE. was achieved. We conclude that homogeneity of the Cu NPs formed on $TiO_2$ is one of the necessary factors to maximize $CH_4$ selectivity in the $ECO_2R$.

**Supplementary Materials:** The following supporting information can be downloaded at: https://www.mdpi.com/article/10.3390/catal12050478/s1, Figure S1: Rietveld analysis results for XRD pattern for (a) $TiO_2$, (b) $Cu$-$TiO_2$, (c) $Cu$-$TiO_2$-H. The observed diffraction intensities, calculated patterns, and the difference between the observed and calculated intensity are denoted by red plus signs, a green solid line, and a blue solid line, respectively; Table S1: Structural parameters determined by Rietveld profile fitting for an XRD pattern of $TiO_2$, $Cu$-$TiO_2$ and $Cu$-$TiO_2$-H; Figure S2: High-angle annular dark-field scanning transmission electron microscopy (HAADF-STEM) image and EDS mapping images of $Cu$-$TiO_2$; Figure S3: Deconvoluted Ti 2p (A) and O 1s (B) XPS spectra of (a) $TiO_2$, (b) $Cu$-$TiO_2$, and (c) $Cu$-$TiO_2$-H. The observed spectra, fitting curves and calculated patterns, and deconvoluted curves are denoted by circle, solid line, and dashed line, respectively; Table S2: XPS peak positions and phase assignment of the $TiO_2$, $Cu$-$TiO_2$, and $Cu$-$TiO_2$-H samples; Figure S4: Example of a gas chromatogram (below: enlarged view in the region for $CH_4$ and CO) of $H_2$, $CH_4$, and CO obtained by electrochemical reduction of $CO_2$ using $Cu$-$TiO_2$-H catalyst on a Molsieve 5A column channel after chronoamperometry operation of 10 min at 1.8 V vs. RHE; Figure S5: Example of a gas chromatogram (below: enlarged view in the region for $C_2H_4$) of $C_2H_4$ obtained by electrochemical reduction of $CO_2$ using $Cu$-$TiO_2$-H catalyst on a Plot Q column channel after chronoamperometry operation of 10 min at 1.8 V vs. RHE; Figure S6: Example of a High Performance Liquid Chromatography (HPLC) of liquid products obtained by electrochemical reduction of $CO_2$ using $Cu$-$TiO_2$-H catalyst after chronoamperometry operation of 10 min at 1.8 V vs. RHE; Table S3: Electrochemical $CO_2$-to-$CH_4$ performance for studies related to highly dispersed or single-site copper catalysts [4,50–53].

**Author Contributions:** Methodology, validation, formal analysis, investigation, data curation, writing—original draft preparation, writing—review and editing, A.A.; Validation, investigation, data curation, M.-H.L.; Methodology, validation, formal analysis, investigation, data curation, K.U.; Validation, investigation, data curation, T.G.N.; Validation, investigation, data curation, A.Y.; Methodology, validation, formal analysis, investigation, data curation, K.K.; Methodology, validation, formal analysis, investigation, data curation, T.S.; Conceptualization, Methodology, resources, writing—review and editing, supervision, project administration, funding acquisition, M.Y. All authors have read and agreed to the published version of the manuscript.

**Funding:** This research was funded by JSPS KAKENHI (JP12852953 and JP18H05517), JST-CREST (15656567), and a project, "Moonshot Research and Development Program" (JPNP18016), commissioned by the New Energy and Industrial Technology Development Organization (NEDO).

**Institutional Review Board Statement:** Not applicable.

**Informed Consent Statement:** Not applicable.

**Data Availability Statement:** The data are contained within the article or Supplementary Materials.

**Conflicts of Interest:** The authors declare no conflict of interest.

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
