# Peer review of "Cu Modified TiO2 Catalyst for Electrochemical Reduction of Carbon Dioxide to Methane"

_catalysts, doi:10.3390/catal12050478_

Round 1

Reviewer 1 Report

This manuscript should be considered for publication after the consideration of the following questions.

What is the novelty of this paper? These points should be clearly stated and highlighted in the introduction part.

Why did the samples contain brookite TiO2?

Why did the author conduct diffuse reflectance UV-Vis spectra since the samples were not photocatalysts?

Did the authors inspect the signals of Ti3+ in the XPS analysis of Cu-TiO2-H? By the way, Figure 4 should describe more clearly that (a) and (b) curves belonged to Cu-TiO2 and Cu-TiO2-H, respectively.

Where were the description of 5B, 5C, and 5D in the manuscript?

Why was the Faraday efficiency higher than 100%?

Was HCOOH detected by HPLC? How many amounts was the gas phase of HCOOH?

How were the authors define CH4 partial current density? What was the mechanism of Cu-TiO2-H that revealed the highest CH4 partial current density?

How did the authors calculate the 70% selectivity? Why did Fig 5E show similar CH4/(C1+C2) for (b) and (c) samples but different CH4 partial current density?

Author Response

REPLY TO THE REVIEWER’S COMMENTS

We thank the reviewers for their instructive comments. All suggestions from the reviewers are very significant and helpful for us to improve the manuscript. We have revised the paper based on the suggestions and, here, we show our replays to the comments.

Reviewer1

1: What is the novelty of this paper? These points should be clearly stated and highlighted in the introduction part.

Reply: We thank for the comment very much. The most important finding of this paper is the high methane selectivity from CO2. This result originates from homogeneous dispersion of Cu species formed on the TiO2 surface, which is achieved via a one-pot preparation of Cu-TiO2 precursors and successive reduction of the precursor under flowing H2. Thus, we revised the manuscript as follows:

Revision [Page 2, lines 50–53]

The increase of surface area is found to efficiently suppresses the Cu agglomeration on a support material such as carbons [6] and oxides [4][7][8]. However, the formation relatively large Cu portions where C–C couplings preferentially progress cannot be suppressed in conventional impregnation synthesis.

2: Why did the samples contain brookite TiO2?

Reply: It was reported that the pH is one of the important parameters for the formation of brookite TiO2, and TiO2 containing brookite phase is synthesized in alkaline condition via hydrothermal/solvothermal synthesis. In this study, we prepared TiO2 and Cu-TiO2catalysts via a solvothermal method using DMF as an alkalescent organic solvent, which may also result in the formation of brookite TiO2, to some extent.

Revision [Page 2, lines 71–74]:

“The intensity of the brookite peak of Cu-TiO2 seemed slightly large compared to that in the XRD pattern of TiO2. It has been reported that the formation of brookite phase preferentially occurs in alkaline conditions. The usage of DMF as an alkalescent organic solvent probably induces the formation of brookite phase [24][25][26].”

3: Why did the author conduct diffuse reflectance UV-Vis spectra since the samples were not photocatalysts?

Reply: Cu complexes and Cu NPs show characteristic optical properties, which are dependent on oxidation states of Cu species and particle sizes, respectively. In this study, we investigated the oxidation state of Cu on Cu/TiO2 catalysts and also the existence of Cu NPs by conducting diffuse reflectance UV-Vis spectra.

Revision [Page 3, lines 95–96]:

“To confirm the oxidation state of Cu on Cu/TiO2, the diffuse reflectance UV-Vis spectra of the catalyst were measured (Figure 2).”

4: Did the authors inspect the signals of Ti3+ in the XPS analysis of Cu-TiO2-H? By the way, Figure 4 should describe more clearly that (a) and (b) curves belonged to Cu-TiO2 and Cu-TiO2-H, respectively.

Reply: We inspected the signals of Ti3+ in the XPS analysis for Cu-TiO2-H. However, the spectrum of Cu-TiO2-H gave a symmetrical peak located around 458.6 eV, which is originated from typical Ti4+ ions contained in anatase TiO2, which was well reproduced by considering only Ti4+, as shown in Figure S2B. Then, we concluded that Ti ions included in the Cu-TiO2-H catalyst exist mainly as a Ti4+ ion, although we could not completely deny the existence of tiny amount of Ti3+. Thus, we revised the manuscript as follows.

Revision [Page 5, lines 144–150]:

“Figure S3A shows Ti 2p XPS spectra of TiO2, Cu-TiO2 and Cu-TiO2-H. All catalysts exhibited a symmetrical peak centered around 458.6 eV, which is a typical peak with characteristic binding energy value for Ti4+ ions contained in anatase TiO2 [33], although the spectrum of TiO2 had a slightly extended tail which possibly comes from the formation of Ti3+. There was also no obvious peak shift in these catalysts. Therefore, Ti ions near the surface of all catalysts would have an analogous chemical environment.”

5: Where were the description of 5B, 5C, and 5D in the manuscript?

Reply: We updated the caption numbers from 5B, 5C, and 5D to 6B, 6C, and 6D in the manuscript. Thank you for pointing that point.

6: Why was the Faraday efficiency higher than 100%?

Reply: The measured total Faradaic efficiencies could sometimes be slightly higher than 100% due to the experimental errors introduced by GC detection or inconsistencies in the flow rate at the outlet. The phenomena were sometimes observed in previous studies. Thus, we revised the manuscript as follows.

Revision [Page 8, lines 205–208]:

“The measured total Faradaic efficiencies on the TiO2 and Cu-TiO2 sometimes became slightly higher than 100%, which has been explained by the experimental errors introduced by GC detection or inconsistencies in the flow rate, as shown in some reports [49][50][51].”

7: Was HCOOH detected by HPLC? How many amounts was the gas phase of HCOOH?

Reply: HCOOH was only detected by HPLC. The gas phase of HCOOH was not detected by online gas chromatography.

Revision [Page 8, lines 197–201]:

“All catalysts produced H2, CO, CH4, and C2H4 as gaseous products, which were detected by online gas chromatography. We could not observe other gaseous product such as HCOOH. Only HCOO was detected in the liquid phase by HPLC analysis as shown in Figure S4-6.”

8-1: How were the authors define CH4 partial current density?

Reply: CH4 partial current density was defined as a product of the average total current density and the Faradaic efficiency for the production of CH4.

Revision [Page 8, lines 210–213]:

“Faradaic efficiency for the production of CH4 reached 18% with 36 mA cm−2 of partial current density at −1.8 V vs. RHE (Figure 6F), where CH4 partial current density was defined as a product of the average total current density and the Faradaic efficiency for the production of CH4 in ECO2R at each potential.”

8-2: What was the mechanism of Cu-TiO2-H that revealed the highest CH4 partial current density?

Reply: The high surface area of Cu species formed on Cu-TiO2-H compared to that on Cu-TiO2 was clarified by STEM observation. The Cu-TiO2-H catalyst possesses a large number of metallic Cu sites on its surface and exhibits high activity for the production of CH4. The band gap of Cu-TiO2-H was narrower than that of Cu-TiO2, indicating that the chemical interaction between Cu and TiO2 in Cu-TiO2-H is well developed than that on Cu-TiO2, which possibly reflect the improvement of electrical conductivity. In summary, high surface area of Cu sites and the improvement of electrical conductivity could be the main reason for the highest CH4 partial current density on the Cu-TiO2-H catalyst.

Revision [Page 8, lines 221–229]:

“Cu-TiO2 and Cu-TiO2-H showed different CH4 partial current densities (Figure 6F (b) and (c)), although the selectivity for CH4 in the products from CO2, i.e., FECH4/FEC1+C2 over these catalysts looked similar (Figure 6E (b) and (c)). DRS results represented that the band gap of Cu-TiO2-H is narrower than that of Cu-TiO2 (Figure 2), which implies that Cu species on Cu-TiO2-H have better contact with the TiO2 support than those on Cu-TiO2. Such favorable interactions between Cu catalysts and the TiO2 support in Cu-TiO2-H may enhance the electrical conductivity, resulting in its high CH4 partial current density on Cu-TiO2-H. These results indicate that Cu(0) loading on TiO2 is indispensable and effective to enhance activity in the ECO2R to CH4.”

9: How did the authors calculate the 70% selectivity? Why did Fig 5E show similar CH4/(C1+C2) for (b) and (c) samples but different CH4 partial current density?

Reply: The 70% selectivity was obtained by calculating the ratios of FE for CH4 to total FE for all products in ECO2R on the catalyst.The similar FECH4/FEC1 + C2values obtained on Cu-TiO2 and Cu-TiO2-H can be attributed to analogous Cu dispersion. In contrast, the different CH4 partial current density for Cu-TiO2 and Cu-TiO2-H samples comes from the improved electrical conductivity of Cu-TiO2-H as expected from the DRS results.

Revision [Page 8, lines 215–229]:

”To evaluate the selectivity for CH4 formation, we further calculated the ratios of FE for CH4 to total FE for all products in ECO2R (FECH4/FEC1 + C2) on the catalysts at each potential (Figure 6E). Cu-TiO2 and Cu-TiO2-H showed larger FECH4/FEC1 + C2 values than TiO2. This result is probably attributable to Cu sites homogeneously dispersed on these catalysts. We achieved 70% of FECH4/FEC1 + C2at −1.8 V vs. RHE, which compares with prior reports related to highly dispersed or single-site copper catalysts (Table S3). On the other hand, Cu-TiO2 and Cu-TiO2-H showed different CH4 partial current densities (Figure 6F (b) and (c)), although the selectivity for CH4 in the products from CO2, i.e., FECH4/FEC1+C2 over these catalysts looked similar (Figure 6E (b) and (c)). DRS results represented that the band gap of Cu-TiO2-H is narrower than that of Cu-TiO2 (Figure 2), which implies that Cu species on Cu-TiO2-H have better contact with the TiO2 support than those on Cu-TiO2. Such favorable interactions between Cu catalysts and the TiO2 support in Cu-TiO2-H may enhance the electrical conductivity, resulting in its high CH4 partial current density on Cu-TiO2-H. These results indicate that Cu(0) loading on TiO2 is indispensable and effective to enhance activity in the ECO2R to CH4.”

Reviewer 2 Report

The manuscript entitled “Cu modified TiO2 catalyst for electrochemical reduction of carbon dioxide to synthesize methane” by Anzai et al. have described the synthesis of a Cu-TiO2 material for electrocatalytic reduction of CO2.  The authors should address the following points to improve the current manuscript.

  1. The introduction part should be improved with few recent examples, and why these metals are important?
  2. The image quality should be improved. The TEM image is not showing the claimed Cu clusters. The elemental mapping from FESEM can be helpful.
  3. The experimental proof for Cu-clusters should be provided.
  4. The mechanistic process should be supported by the experimental data.
  5. How did the authors determine the identity of reduced product e.g. CH4? The characteristic analysis (e.g. NMR) should be provided for the products.
  6. The authors should provide the stability and post-catalysis characterizations of the material.
  7. A comparative study with previous reports should be provided to establish the novelty of this work.

Author Response

REPLY TO THE REVIEWER’S COMMENTS

We thank the reviewers for their instructive comments. All suggestions from the reviewers are very significant and helpful for us to improve the manuscript. We have revised the paper based on the suggestions and, here, we show our replays to the comments.

Reviewer2

1: The introduction part should be improved with few recent examples, and why these metals are important?

Reply: We thank the reviewer’s comments to improve this paper. According to the comment, we revised the introduction.

Revision [Page 2, lines 47–53]:

“Recently, the formation of isolated Cu sites has been found to effectively improve the selectivity for the CH4 production by suppressing the unfavorable C–C coupling [4][5]. However, the selectivity for the production of CH4 from CO2 (CO2 to CH4 selectivity) should be improved. The increase of surface area is found to efficiently suppresses the Cu agglomeration on a support material such as carbons [6] and oxides [4][7][8]. However, the formation relatively large Cu portions where C–C couplings preferentially progress cannot be suppressed in conventional impregnation synthesis.”

2: The image quality should be improved. The TEM image is not showing the claimed Cu clusters. The elemental mapping from FESEM can be helpful.

Reply: Thank you for the suggestion. We replaced the Figure 3 with more clear STEM images with EDS mapping images.

3: The experimental proof for Cu-clusters should be provided.

Reply: We replaced the Figure 3 with more clear STEM images with EDS mapping images to provide the experimental proof for Cu-clusters.

4: The mechanistic process should be supported by the experimental data.

Reply: According to the results shown in Figure 6, HER dominantly proceeded on the bare TiO2 with producing small amount of C1 products, whereas selectivity to ECO2R was improved on the Cu-TiO2 and Cu-TiO2-H. This indicates that the inclusion of Cu on TiO2would play a key role in the formation of CH4. Therefore, the mechanism of ECO2R on Cu-TiO2 might be similar to that on Cu electrodes, as reported so far. Therefore, homogeneous dispersion of Cu species on the TiO2 would contribute to the high CH4selectivity.

Revison [Page 10, lines 239–252]:

“It has been reported that composite materials of Cu and CeO2 exhibit high CH4 selectivity [4][7][44][45], where abundant oxygen vacancy (VO) sites on CeO2 play an important role in both CO2 adsorption and activation. Wang et al. suggested that Cu site surrounded by 3VO effectively promoted CH4 formation, where the 2VO neighboring Cu are filled with the two oxygen atoms of CO2 and should enhance the CO2 adsorption and activation [4]. In this study, the surface of the prepared TiO2 samples may consist of low concentration of VO. CO2 dissociation unfavorably occur on stoichiometric anatase surface compared to Vo-rich surface due to its large CO2dissociation energy [46]. HER mainly proceeded on the TiO2 catalyst, although small amount of C1 products, such as CO, HCOO, and CH4, were generated (Figure 6B). In contrast, Cu-TiO2 and Cu-TiO2-H, where Cu(0) species were well dispersed over a TiO2 grain, showed high selectively for the production of CH4. Hence, well dispersed Cu species on TiO2 would play a key role in the formation of CH4. Therefore, it is reasonable that homogeneity of Cu NPs in their size and dispersity on TiO2 can be a key for the high CO2-to-CH4selectivity.”

5: How did the authors determine the identity of reduced product e.g. CH4? The characteristic analysis (e.g. NMR) should be provided for the products.

Reply: The products in ECO2R were identified by an online gas chromatography for gaseous products, such as H2, CO, CH4, C2H4, and by HPLC for liquid products such as HCOOH. According to the comment, we added the chromatogram images of gas chromatography and HPLC analysis for the products in the supporting information (Figures S4-S6).

Revision [Page 8, lines 197–201]:

“All catalysts produced H2, CO, CH4, and C2H4 as gaseous products, which were detected by online gas chromatography. We could not observe other gaseous product such as HCOOH. Only HCOO was detected in the liquid phase by HPLC analysis as shown in Figure S4-6.”

6: The authors should provide the stability and post-catalysis characterizations of the material.

Reply: In this study, we conducted the ECO2R using a flow cell where the catalyst is loaded onto a traditional carbon-based gas diffusion electrode (GDE) because of its superior mass transportation property. However, the stability of the carbon-based GDE was poor and damaged or corrupted at the worst case within 40-60 min operation, and then, a selectivity to ECO2R decreased and HER dominantly proceeded. The low stability of this carbon-based GDE is caused by rapid GDE flooding during ECO2R, encouraging salt precipitation [Dinh, C.T.; Burdyny, T.; Kibria, M.G.; Seifitokaldani, A.; Gabardo, C. M.; García De Arquer, F.P.; Kiani, A.; Edwards, J.P.; De Luna, P.; Bushuyev, O.S.; Zou, C.; Quintero-Bermudez, R.; Pang, Y.; Sinton, D.; Sargent, E.H. CO2 electroreduction to ethylene via hydroxide-mediated copper catalysis at an abrupt interface. Science, 2018, 360, 783-787. doi:10.1126/science.aas9100]. To solve the problems of the poor stability of carbon-based GDE, recently novel types of GDEs composed of metal, PTFE, and membrane have been developed. Therefore, we need to prevent the deterioration of the GDE before the evaluation of the Cu/TiO2 catalysts, which is currently underway.

7: A comparative study with previous reports should be provided to establish the novelty of this work.

Reply: According to the comment, we add Table S3 in the Supporting information for comparative study with previous reports and the following sentence in our manuscript as below.

Revision [Page 8, lines 219–221]:

“We achieved 70% of FECH4/FEC1 + C2 at −1.8 V vs. RHE, which compares with prior reports related to highly dispersed or single-site copper catalysts (Table S3).”

Reviewer 3 Report

Chennai,

16th March, 2022.

Manuscript ID: Catalysts-1654447

Title: Cu modified TiO2 catalyst for electrochemical reduction of carbon dioxide to synthesize methane.

Authors: Akihiko Anzai, Liu Ming-Han, Kenjiro Ura, Tomohiro G Noguchi, Akina Yoshizawa, Kenichi Kato, Takeharu Sugiyama and Miho Yamauchi*

Summary:  The original paper of Yamauchi et al., deals with the solvothermal synthesis of Cu species supported titania (anatase) catalysts and using the same for the electrochemical reduction of CO2 to methane.  Three catalysts were studied for this purpose.  TiO2 (anatase) as such, copper acetated supported TiO2 upon thermal treatment (CuxO/TiO2) and copper acetated supported TiO2 upon reduction in hydrogen environment (Cu/TiO2 designated as Cu-TiO2-H).  Among the three catalysts, Cu-TiO2-H showed remarkable activity and selectivity for the electrochemical reduction of CO2 to CH4.  The outstanding performance of Cu-TiO2-H was attributed to the small particle size of the Cu nanoparticles (2-3 nm) that are homogeneously distributed on the TiO2 surface.   With the Cu-TiO2-H catalyst, the Faradaic efficiency for methane production from the electrochemical reduction of CO2 reached 18% with a partial current density of 36 mAcm-2 at -1.8 V vs RHE.  70% selectivity for CH4 was observed at -1.8 V vs RHE.  Though a web of Science search with the keywords, namely, CO2 electrochemical reduction and CH4 and Cu and TiO2, showed one result, it did not match with the findings reported by Yamauchi et al., Owing to the originality and usefulness of the results, the work of Yamauchi et al., is recommended for publication in the journal Catalysts after minor revision. 

Major issues: None

Minor issues:   English language needs improvement.  Spelling mistakes and grammatical errors and improper spacing between words and reference numbers and improper citing of the name of Chinese authors needs correction.  The following specific changes can be made for improving the quality of the paper.

Line number

Revision

3

Delete “synthesize”

53

Insert “space” between “application” and “[6]”

338

Replace “CO2” with “CO2

Apply this change throughout the paper

Subscripts and superscripts to be properly represented

342

Replace “CO2” with “CO2

345

Add “?” after “fuels”

347

Replace “CH4” with “CH4

346

Replace “Wang, Y.; Chen, Z.; Han, P.; Du, Y.; Gu, Z.; Xu, X.; Zheng, G.” with “Wang, Y. F.; Chen, Z.; Han, P.; Du, Y. H.; Gu, Z.X.;”

349

Replace “CO2” with “CO2

348

Replace “Xu, Y.; Li, F.; Xu, A; Edwards, J. P.; Hung, S. F.; Gabardo, C. M.; O’Brien, C. P.; Liu, S.; Wang, X.; Li, Y.;” with “Xu, Y.; Li, F. W.; Xu, A. N.; Edwards, J. P.; Hung, S. F.; Gabardo, C. M.; O’Brien, C. P.; Liu, S. J.; Wang, X.; Li, Y. H.; Wicks, J.; Miao, R. K.; Liu, Y.; Li, J.; Huang, J. E.; Abed, J.; Wang, Y. H.; Sargent, E. H.; Sinton, D.;”

351

Replace “TiO2” with “TiO2

353

Replace “CO2” with “CO2

356

Replace “TiO2” with “TiO2

372

Replace “TiO2” with “TiO2

382

Replace “TiO2” with “TiO2

384

Replace “TiO2” with “TiO2

389

Replace “TiO2” with “TiO2

419

Replace “CO2” with “CO2

419

Replace “CH4” with “CH4

421

Replace “CO2” with “CO2

421

Replace “TiO2” with “TiO2

427

Replace “.” After “Electrocatalyst” with “?”

40

Replace “sufficiently achieved” with “achieved to the desired level”

51

Add “is” before “ubiquitous”

53

Insert “space” between “application” and “[6]”

53

Replace “has” with “have”

58

Replace “deribed” with “derived”

61

Add “by the electrochemical reduction of CO2” after “CH4 production”

68

Delete “origin”

69

Replace “in” with “of”

69

Replace “Cu2O” with “Cu2O”

74

Delete “contained”

75

Replace “CuOx” with “CuOx

78

Replace “Rietvelt” with “Rietveld”

81

Replace “were found to consist of” with “consisted of”

83

Replace “Rietvelt” with “Rietveld”

84

“1.9 % out of 10 %” is not large.  Why does the authors say “a large amount of Cu species was deposited”.  What happened to the rest of the 8.1 % Cu precursor?

90

Replace “to the” with “as a function of”

91

Replace “x-intercept” with “X-intercept”

109

Figure 2: replace “R∞” with “R

105

Replace “on” with “in”

106

Replace “XPS” with “DRS”

114

Replace “dispersity” with “dispersion”

114

Add “electron” after “transmission”

118

Replace “on” with “in”

112

Delete “a” before “TiO2

128

Insert “space” between “Cu(OH)2” and “[25]”

132

Replace “as mainly” with “mainly as”

138

Add “characteristic” before “binding energy”

139

Replace “among” with “in”

155

The binding energy values and their attribution to a particular species in the catalysts studied can be summarized in a tabular form for easy comprehension

143

Replace “includes” with “contained”

144

Insert “space” between “530.8” and “eV”

132

Replace “as mainly” with “mainly as”

133

Replace “attributable” with “attributed”

138

Add “value characteristic of” after “energy”

139

Replace “shift among” with “shifts in”

143

Replace “includes” with “contains”

145

Add “peak” after “other”

152

Add “by the” before “change”

164

Add “absorption” after “X-ray”

161

Replace “X-ray absorption fine structure spectra (XAFS)” with “X-ray absorption near edge structure (XANES) spectra”

161-162

Delete “for the Cu-TiO2 and Cu-TiO2-H”

164

Add “absorption” after “X-ray”

165

Replace “(XANES)” before “spectra”

165

Replace “of” before Cu-TiO2-H with “and”

166

Replace “Cu2O” with “Cu2O”

167

Replace “assignable” with “attributed”

167

Replace “an” with “is”

170

Delete “characterized with”

171

Replace “was analysed in more detail” with “were further analysed in detail”

171

Replace “spectra” with “spectrum”

177

Replace “can be” with “was”

184

Add “XANES” before “spectra”

184

Add “(A)” after “Cu-K-edge spectra”

184

Add “(B)” after “first derivative of Cu-K-edge spectra”

193

Replace “was” with “were”

194

Replace “CO2 selectivity” with “activity for CO2 electro reduction”

213

Add “that” before “composite”

216

Replace “to” with “on”

219

Insert “space” between “energy” and “[36]”

220

Replace “product” with “products”

224

Replace “ECO2RR” with “ECO2R”

232

Replace “pod” with “pot”

234

Replace “CuOx” with “CuOx

236

Replace “toward” with “for”

241-336

For want of time these lines could not be read line by line but were only glanced through.  The authors are advised to correct for scientific/English errors if any to avoid another revision.

Author Response

REPLY TO THE REVIEWER’S COMMENTS

We thank the reviewers for their instructive comments. All suggestions from the reviewers are very significant and helpful for us to improve the manuscript. We have revised the paper based on the suggestions and, here, we show our replays to the comments.

Reviewer3

We thank the reviewer for his/her careful checking the manuscript. According to the comment, we check throughout the paper, and corrected the typos, grammatical errors, improper spacing between words and reference numbers and improper citing of the name of authors. Thank you very much for pointing them out.

84: “1.9 % out of 10 %” is not large. Why does the authors say “a large amount of Cu species was deposited”. What happened to the rest of the 8.1 % Cu precursor?

Reply: According to the comment, we reconsidered the XRD results. Rietveld profile fitting of the XRD patterns shows that the weight fraction of fcc Cu precipitates is 1.9% and the remaining Cu species are incorporated into the TiO2 lattice or exist as amorphous-like Cu species. We compared the lattice constants of TiO2 in detail, we found that they have slight differences, suggesting that Cu was partially incorporated into the TiO2 lattice. Since the change in the lattice constant is slight, we cannot rule out the possibility that amorphous Cu species also present. Thus, we revised our manuscript as below.

Revision [Page 2, lines 82–91]:

”To obtain detailed structural parameters for the catalysts, We conducted Rietveld profile fitting of these XRD patterns (Figure S1). Structural parameters are summarized in Table S1. The lattice constants of anatase phase constituting Cu-TiO2 and Cu-TiO2-H showed a slight increase in the a-axis but a slight decrease in the c-axis, compared to those of pure TiO2, suggesting the possibility of incorporation of Cu species into the TiO2 lattice. The weight fraction of Cu species deposited on TiO2 was estimated by the Rietveld analysis to be low (1.9%) even though the initial starting amount was 10%. Considering the slight change in the structure of Cu-TiO2and Cu-TiO2-H, relatively large percentage of Cu species (more than 5%) possibly exist as amorphous on the surface of TiO2.”

155: The binding energy values and their attribution to a particular species in the catalysts studied can be summarized in a tabular form for easy comprehension.

Reply: According to the comment, we add Table S2 in the Supporting information to summarize the binding energy values and their attribution to a particular species in the catalysts and the following sentence in our manuscript as below

Revision [Page 6, lines 160–163]:

“The amount of defects such as Ti3+ and oxygen vacancy on the surface of these catalysts did not change so much by the introduction of Cu or by the change of the atmosphere during heating. The all measured core level positions for the samplesare summarized in Table S2.”
